# Anti-Arthritic and Anti-Inflammatory Potential of *Spondias mangifera* Extract Fractions: An In Silico, In Vitro and In Vivo Approach

**DOI:** 10.3390/plants10050825

**Published:** 2021-04-21

**Authors:** Mohammad Khalid, Mohammed H. Alqarni, Ambreen Shoaib, Muhammad Arif, Ahmed I. Foudah, Obaid Afzal, Abuzer Ali, Amena Ali, Saad S. Alqahtani, Abdulmalik S. A. Altamimi

**Affiliations:** 1Department of Pharmacognosy, College of Pharmacy, Prince Sattam Bin Abdulaziz University, P.O. Box 173, Al-Kharj 11942, Saudi Arabia; m.alqarni@psau.edu.sa (M.H.A.); a.foudah@psau.edu.sa (A.I.F.); 2Department of Clinical Pharmacy, College of Pharmacy, Jazan University, Jazan 45142, Saudi Arabia; asahmad@jazanu.edu.sa (A.S.); ssalqahtani@jazanu.edu.sa (S.S.A.); 3Department of Pharmacognosy, Faculty of Pharmacy, Integral University, Lucknow 226026, India; arifxyz@iul.ac.in; 4Department of Pharmaceutical Chemistry, College of Pharmacy, Prince Sattam Bin Abdulaziz University, P.O. Box 173, Al-Kharj 11942, Saudi Arabia; obaid263@gmail.com (O.A.); as.altamimi@psau.edu.sa (A.S.A.A.); 5Department of Pharmacognosy, College of Pharmacy, Taif University, P.O. Box 11099, Taif 21944, Saudi Arabia; abuali@tu.edu.sa; 6Department of Pharmaceutical Chemistry, College of Pharmacy, Taif University, P.O. Box 11099, Taif 21944, Saudi Arabia; amrathore@tu.edu.sa

**Keywords:** arthritis, anti-inflammatory, *Spondias mangifera*, in silico, in vitro and in vivo

## Abstract

The fruits of *Spondias mangifera (S. mangifera)* have traditionally been used for the management of rheumatism in the northeast region of India. The present study explores the probable anti-arthritis and anti-inflammatory potential of *S. mangifera* fruit extract’s ethanolic fraction (EtoH-F). To support this study, we first approached the parameters in silico by means of the active constituents of the plant (beta amyrin, beta sitosterol, oleonolic acid and co-crystallised ligands, i.e., SPD-304) via molecular docking on COX-1, COX-2 and TNF-α. Thereafter, the absorption, distribution, metabolism, excretion and toxicity properties were also determined, and finally experimental activity was performed in vitro and in vivo. The in vitro activities of the plant extract fractions were evaluated by means of parameters like 1,1-Diphenyl-2- picrylhydrazyl (DPPH), free radical-reducing potential, albumin denaturation, and protease inhibitory activity. The in vivo activity was evaluated using parameters like COX, TNF-α and IL-6 inhibition assay and arthritis score in Freund Adjuvant (CFA) models at a dose of 400 mg/kg b.w. per day of different fractions (hexane, chloroform, alcoholic). The molecular docking assay was performed on COX-1, COX-2 and TNF-α. The results of in vitro studies showed concentration-dependent reduction in albumin denaturation, protease inhibitors and scavenging activity at 500 µg/mL. Administration of the *S. mangifera* alcoholic fraction at the abovementioned dose resulted in a significant reduction (*p* < 0.01) in arthritis score, paw diameters, TNF-α, IL-6 as compared to diseased animals. The docking results showed that residues show a critical binding affinity with TNF-α and act as the TNF-α antagonist. The alcoholic fraction of *S. mangifera* extract possesses beneficial effects on rheumatoid arthritis as well as anti-inflammatory potential, and can further can be used as a possible agent for novel target-based therapies for the management of arthritis.

## 1. Introduction

In ancient times, traditional systems of medicine were the fundamental source of herbal medications [1]. A majority of the population is dependent the on use of various species of herbal remedies to treat health problems [2] because of the insufficient availability of modern medicine, particularly in rural areas [3].

Rheumatoid arthritis (RA) is an autoimmune disorder, which can result in chronic inflammation in the synovial membrane and also cause pain in small and large joints, as well as the destruction of cartilage and bone [4]. The characteristic features of RA are joint pain, immobility and malformation [5]. The management of RA is mainly achieved through the use of nonsteroidal anti-inflammatory drugs (NSAIDS) such as indomethacin, ibuprofen, aspirin, and naproxen, but these only manage it for a short time duration [6]. The arthritic and anti-inflammatory action of NSAID is attributed to it cyclooxygenase (COX-1 and 2) inhibition, as well as its inhibition of the pro cytokinins (IL1, IL-6 TNF-alpha, etc.), curing arthritic disease [7]. Some NSAID has a short duration of action and can also produce some negative side effects in the epigastric region [7]. When inflammation occurs, macrophage cells are released into the injured tissue area, which can cause life-threatening diseases like Alzheimer’s, arthritis, cancer, allergies, and atherosclerosis, as well as autoimmune diseases [1]. Inflammation causes the vasodilation of capillaries and increases the blood flow to the injured region [8].

Oxidation causes chemical and physiological changes in the biological system in living organisms and organic substances, which can be oxidised as a result of various physicochemical processes such as exposure to heat, light or any other oxidising agents [9]. Reactive oxygen species (ROS) are produced from the oxidised bioactive constituents, as a result of the direct exposure of highly reactive molecules that are abundant in living tissue to the atmosphere or during aerobic metabolism [10]. ROS circulating in the blood stream affects the metabolic process because it reacts with the free electron molecules that are present in living systems, which can lead to various life-threating diseases like ischemia, respiratory distress, arthritis, cancer, and aging, and can also damage various vital organs. Herbal medicines have numerous bioactive phytoconstituents, and have the ability to quench free radical oxygen species; this contributes beneficial effects towards life-threatening diseases in our body [11].

*Spondias mangifera* Willd. (Anacardiaceae), known as the bile tree, in Ayurveda system is also known as Amrata. It is widely distributed in the tropical and northeast region of India, and is cultivated in several states, but mainly in Punjab, Maharashtra, Bengal [12]. In ancient times, in the north-east region of India, people used it for the treatment of rheumatism [13]. Like various herbs, it has an odour that resembles that of turpentine upon scrubbing and/or chewing. It has numerous bioactive phytoconstituents in various parts of the plant: the fruits and aerial parts contain cycloartanone-24-methylene, daucosterol, lignoceric acid, stigmast-4-en-3-one, cystine, oleanolic acid and *β*-amyrin; and glycine and leucine respectively [14]. The fruits also contain some phytoconstituents like alanine, galloylgeranin, vit-A, riboflavin, and niacin in very small amounts [15].

The green fruit is useful in biliousness and dyspepsia, while the fruit powder also has anti-tubercular properties, and can also be used as an astringent, refrigerant, or tonic, and is used for the management of rheumatic arthritis and myalgia [12]. In indigenous systems of medicine, bark is used as a rubefacient on the skin over painful joints, and bark paste is used as an embrocation for both articular and muscular rheumatism [13]. In the Hazaribag district, bark paste with garlic is used in the stomach [16]. The roots are used in treatments for regulating menstruation, and possess antibacterial [17], antitumor [18], antispasmodic and antihistamine activities, and decoctions of root bark are used in the treatment of gonorrhoea. The methanolic extracts of stem heart wood possess hepatoprotective activity against carbon tetrachloride-induced liver impairment in rodents [19]. The juice of the ripe fruit of this plant is a rich source of several vitamins and can potentially be used as a nutraceutical agent [20]. The fruits and barks of the plant are also used in diabetes [21].

The present study was designed to determine the pharmacokinetics and pharmacodynamic properties of *Spondias mangifera.* Among all of the phytochemicals, β sitosterol, β amyrin and oleonolic acid are active. Bearing in mind the special considerations for determining the anti-inflammatory and anti-arthritic actions of the plant, first, the active constituents were tested in a docking study to determine the binding affinity, and then they were further tested in a pharmacokinetics study. Finally, the ethanolic fraction of the plant extract was used for the estimation of pharmacological activity against inflammation and CFA-induced arthritis in an animal model. 

## 2. Materials and Methods

### 2.1. Plant Collection and Authentication

Crude fruits of the plant were collected from the local market in Aminabad, Lucknow India. The fruits were identified by the taxonomist at the Department of Pharmacognosy, faculty of Pharmacy, Integral University, Lucknow. A voucher sample was submitted to the herbarium for supplementary reference (IU/PHAR/HRB/20/16).

### 2.2. Chemicals

Ascorbic acid, ferrous sulphate, aluminium chloride, sulphanilamide, phosphoric acid, naphthylene diamine dihydrochloride, and rutin were bought from Qualigens Fine Chemicals, Mumbai, India. DPPH and folin ciocalteu reagents were purchased from Sigma-Aldrich, St. Louis, USA. Sodium carbonate, gallic acid and trypsin were attained from Merck Chemicals (Darmstadt, Germany). Tris-HCl buffer, Ferrous ammonium sulphate, and sodium dodecyl sulphate (Qualigens Fine Chemicals, Mumbai, India). All the reagents used in the experiment were of analytical grade.

### 2.3. Molecular Docking

Glide 5.9, executed in Maestro 9.4 (GUI of Schrodinger), was used for extra-precision (XP) docking of β-amyrin, β-sitosterol, oleonolic acid and co-crystallised ligands. The X-ray crystal structures of COX-1 (PDB ID: 3N8Z, resolution: 2.90 Å), COX-2 (PDB ID: 4PH9, resolution: 1.81 Å), and TNF-α (PDB ID: 2AZ5, resolution: 2.10 Å) were obtained from Protein Data Bank (PDB) and used for in silico study [22,23,24]. Protein Preparation Wizard in Maestro was employed for the protein structure preparation, viz., omission of water particles, required bond orders, enclosure of hydrogen atoms, and treatment of formal charges. A comprehensive sampling option was employed for the optimisation of the hydrogen bonding grid. The energy of the protein structures was minimised to an RMSD of 0.3 Å using the impref module (Impact 5.9) with the OPLS_2005 force field. Glide scoring grids (docking grid box of 20 × 20 × 20 Å) were produced using the active binding site residues in the protein structure. LigPrep 2.6 and Epik 2.4 were used to expand protonation, as well as tautomeric states of β-amyrin, β-sitosterol, oleonolic acid and co-crystallised ligands at pH 7.0 ± 2.0, and then the energy was decreased by means of the OPLS_2005 force field. The docking simulation of the set ligands was performed by means of Glide XP docking [25].

### 2.4. Prediction of Swiss Absorption, Distribution, Metabolism, Excretion and Toxicity Properties (ADME/Tox) of the phytoconstituents from Spondias mangifera

Prediction of ADME properties was carried out using Swiss ADME software from the Swiss Institute of Bioinformatics, accessed via a web server that displays the submission page of the Swiss ADME for estimating the individual ADME behaviours of the prominent compounds of *S. mangifera* (gallic acid, kaempferol, quercetin and ascorbic acid). The list was constructed such that it contained one input molecule per line with several inputs, defined by a simplified molecular input line entry system, and the results for each molecule were presented in the form of tables, graphs and an excel spreadsheet. The significant ADME properties predict both physicochemically significant descriptors and pharmacokinetically relevant properties. ADME properties determine the drug-like activity of ligand molecules on the basis of Lipinski’s rule of five. In this study, they were used to measure the safety of the compounds present in the plant *S. mangifera* [26].

### 2.5. Extract and Fraction Preparation

The *S. mangifera* fruits were collected and dried; for the drying process, they were scattered on a dry open area at room temperature. After air drying, the fruits were cut into four pieces by the means of a sharp knife. In addition, they were again dried in an oven at 40–45 °C for a period of 2–3 days until constant weight. The dried fruits were made into coarse powder using a grinder, and the powder materials were defatted with petroleum ether and further macerated with methanol for 72 h with occasional shaking. The extract was filtered with Whatmann filter paper three times at intervals of one day the extract was kept on a rotatory evaporator at low temperature to obtain a viscous and sticky mass of the extract.

The filtrate was fractionated through the solvent according to ascending order of polarity, as follows: hexane < chloroform < ethanol respectively. Each fraction was concentrated up to dryness at 40 °C by using a rotary evaporator. The dried fractionated extract was then kept at 4 °C until further use [11].

### 2.6. Animals

Albino swiss mice (both sexes) of equal weight (35 g) were purchased from the National Laboratory Animal centre, Central Drug Research Institute, Lucknow, Uttar Pradesh, India (Approval No.: IU/CPCSEA/08/ac/1213). The necessary approval from the Institutional Ethical Committee was obtained, and fresh and healthy animals were selected. No experimental study was conducted. The animals were housed separately in polypropylene cages at a temperature of 21 + 2 °C and relative humidity (55 ± 5%), with a light/dark cyclic of 12-h, for one week and fed with pellets and water ad libitum.

### 2.7. COX Inhibition Assay

This test was performed as per the Viji and Helen assay. According to the given protocol, a mixture was prepared using tris-HCl buffer enzyme, haemoglobin, glutathione, arachidonic acid and Trichloroacetic acid (TCA; 10% in 1N HCl, 0.2 mL). This mixture was then incubated for approximately 20 min at 37 °C. Thereafter, 0.2 mL of thiobarbituric acid (TBA reagent) was added to the mixture, which was subsequently kept on boiling water for 20 min. Afterward, it was cooled and centrifuged at 1000 rpm for 3 min. The final supernatant was used for the measurement of COX activity at 560 nm [27].
COX inhibition activity (%) = 1 − T/C × 100

T is the absorbance of the inhibitor well (at 560 nm) and C denotes the initial absorbance activity without the inhibitor well (at 560 nm).

### 2.8. TNF-α and IL-6 Inhibition Assay

The Tumour Necrosis Factor alpha (TNF-α) inhibitory assay of the *S. mangifera* fractions was performed using an ELISA kit (Thermo Scientific, Waltham, USA) for quantification of TNF-α and IL-6 in the experimental rats’ serum. In accordance with the instruction leaflet, absorbance was measured at 450 and 550 nm on an ELISA plate reader.

### 2.9. 1,1-diphenyl-2-Picrylhydrazyl Scavenging Assay

Different fractions of *S. mangifera* were analysed using the method provided in Liyana and Shahidi, with slight modification. DPPH (0.135 mM) solution was mixed in 1 ml of ethanol and kept on vortex to mix thoroughly. The reaction mixture was incubated at a temperature of 25 °C for a time period of approximately 15–20 min, and then further absorbance was recorded at 490 nm. The scavenging activity of DPPH was calculated as per the formula below [28].
% Inhibition = [(A0 − A1)/A0 × 100]
where A0 is the absorbance of the blank, whereas A1 is the absorbance of the extract fraction sample.

### 2.10. Reducing Potential Scavenging Activity

According to the Khalid et al. method, the transformation of Fe^+3^ to Fe^+2^ in the presence of the fractions was measured at an absorbance of 700 nm after adding the regents. Increased absorbance of the reaction mixture indicates increased reducing power. Gallic acid was used as standard [29].

### 2.11. Inhibition of Albumin Denaturation

Egg albumin (0.2 mL) was added in all the fractionated extracts; to these mixtures, phosphate-buffered saline (2.8 mL; pH 6.4) was added. The final volumes of the samples were then made up with distilled water up to 5 mL. For control purposes, double-distilled water and aspirin were used as standard. The fractionated extracts were incubated in a biological oxygen demand chamber at 37 °C for 15 min, following which the mixtures were heated at 70 °C for no longer than 5 min. Then, these mixtures were cooled, and the absorbance was measured at 660 nm [30].

The formula for the calculation of percentage inhibition of protein denaturation is mentioned below:$$Percentage Inhibition\;=\;\frac{Abs control-Abs treated\times 100}{Abs treated}$$

### 2.12. Protease Inhibition Assay

This procedure was performed as per the protocol of Dharmalingam and group, with a slight modification. For this, 2 mL of reaction mixture consisting of trypsin (0.06 mL), Tris HCl buffer (1 mL; 20 mM; pH 7.4) and fractionated extracts of the plant (1 mL) was incubated for 10 min at 37 °C. Afterward, casein (1 mL of 0.65% *w*/*v*) was added, and the mixture was further re-incubated for 20 min. Subsequently Perchloric acid (2 mL; 2 M) was added to that mixture, a cloudy suspension was obtained and centrifuged at 7830 rpm for 20 min. The absorbance of the supernatant liquid was measured at 280 nm. Tris-HCl buffer solution was used as a control. The formula that was used for the calculation of % inhibition is mentioned below [31].
% inhibition = (1 − Ac/At) 100

In this Ac denotes the absorbance of control sample whereas at shows the absorbance related with test sample.

### 2.13. Complete Freund’s Adjuvant (CFA)-Induced Arthritis in Rats

The arthritis model was performed in pathogen-free experimental rats by the plantar injection of 0.1 mL of CFA on day 0. The experimental rats were randomly divided into six groups. Group I served for the normal control (vehicle). Group II and III were CFA and CFA plus Aspirin (100 mg/kg per day p.o.), respectively. Group IV was the treated group of CFA plus hexane extracted fraction (Hx-F; 400 mg/kg per day p.o.). Group V served as CFA plus chloroform extracted fraction (Chl-F; 400 mg/kg per day p.o.). Group VI served as the treated CFA group plus ethanol extracted fraction (EtOH-F; 400 mg/kg per day p.o.). The experiment was performed for 40 days with the respective groups of animals. Animals were examined carefully throughout the experiments [32].

### 2.14. Arthritic Score

The degree and score of arthritis was monitored daily by means of a scale from 0 to 4 for each paw, aiming for a maximum score of 8 per rat. After induction of arthritis, the joint diameters of the right hind paw were measured using an electronic Vernier calliper (Fischer Scientific, CON3417) [33].

The scoring criteria are mentioned below:Normal paw = 0,Mild swelling and erythema of digits = 1,Swelling and erythema of the digits = 2,Severe swelling and erythema = 3,Gross deformity and inability to use the limb = 4.

### 2.15. Statistical Analysis

The results are expressed as mean ± SEM of six animals per group. Statistical significance was calculated by means of ANOVA (Graph Pad Software Inc., San Diego, USA), followed by Dunnet “*t*” test, and *p <* 0.05 was considered statistically significant.

## 3. Results

To the best of our knowledge, this is the first study that includes pharmacokinetics, dynamics, in silico, in vitro and in vivo studies on the extract fraction of the plant *Spondias mangifera*. This shows the probable anti-inflammatory and anti-arthritic action of the plant. The outcomes of the results can be proved with respect to several aspects, described below.

### 3.1. Molecular Docking

The major constituents beta amyrin, beta sitosterol, and oleanolic acid were docked into the active site of COX-1(PDB ID: 3N8Z, resolution: 2.90 Å), COX-2 (PDB ID: 4PH9, resolution: 1.81 Å), and TNF-α (PDB ID: 2AZ5, resolution: 2.10 Å) using Glide XP docking via the Maestro GUI of Schrodinger. None of the compounds showed any binding affinity with COX-1 and COX-2. The major constituents β-amyrin, β-sitosterol, and oleonolic acid showed excellent binding affinity with pro-inflammatory cytokine TNF-α. The co-crystallised ligand SPD-304 showed a similar binding pose to that in the crystal structure, validating our docking protocol. The summary of the Glide XP score and interacting binding residues is provided in Table 1. The 2D and 3D illustrations of the binding pose of beta amyrin, beta sitosterol, oleonolic acid and SPD-304 at the interface of chain A and B of TNF-α are presented in Figure 1 and Figure 2, respectively.

### 3.2. Server-Based ADME Analysis

Server-based ADME analysis provides rapid results, which can be useful for the development of lead compounds. In our study, we used Swissdock for the ADME analysis of selected molecules from the literature. The results of ADME analysis are described in Table 2 and the docking figure is shown in Figure 3. The predicted properties of kaempferol, gallic acid, quercetin and ascorbic acid are within ranges satisfying all of the stipulations of Lipinski’s rule of five for consideration as having drug-like potential. Figure 3 clearly shows that there are no compounds that cross the blood–brain barrier. Among these compounds, compound gallic acid and caffeic show the best ADME properties.

### 3.3. Cyclooxygenase Assay

The COX-1 and COX-2 inhibition effect was investigated using different fractions of *S. mangifera* extract. None of the fractions of *S. mangifera* extract, i.e., hexane, chloroform, ethanol fractions, showed a percentage inhibition of cyclooxygenase (COX 1 and 2) activity. This means that the test fractions and reagents did not react with each other. Therefore, the results for this are not included.

### 3.4. TNF-Alpha

This study revealed that TNF-α plays a significant role in the inflammatory process. The highest concentration of TNF-α resulted in a significant (*p <* 0.01) improvement in the toxic control compared to the normal group. When the ethanolic fraction of *S. mangifera* was orally administered at a dose of 400 mg/kg per day, the concentration of TNF-α showed a significant (*p <* 0.05) reduction (71.83 ± 2.71), whereas the other fractions, like Hx-F (131.72 ± 4.91) and Chl-F (89.51 ± 3.21), did not show a significant (*p >* 0.05) effect on TNF-α concentration when compared with the toxic control group at this dose. Group III shows significantly reduced the concentration of TNF - α when compared with group II (68.51 ± 2.98). (Figure 4). The evaluated concentration of IL-6 was significantly (*p <* 0.01) increased in the group II animals compared to those in group I. Group III shows significantly reduced the concentration of IL - 6 when compared with group II (61.51 ± 2.17). When the alcoholic fraction of *S. mangifera* extract was administered at a dose of 400 mg/kg in the Group VI animals, the results showed a significant (*p <* 0.05) reduction (68.83 ± 1.87) in the IL-6 concentration, whereas there was no reduction in the IL-6 concentration in groups IV and V (121.72 ± 3.71 and 76.51 ± 2.81, respectively) when compared to the group II animals (Figure 5).

### 3.5. DPPH Scavenging Activity

This study showed that the ethanolic fraction of *S. mangifera* has greater scavenging activity than the other fractions (hexane, chloroform, and ethanol) in comparison to the standard (ascorbic acid). It significantly showed a maximum percentage inhibition of 87.01 ± 2.73 mg/mL, while ascorbic acid was 101.72 ± 3.17, at a dose of 1.5 mg/mL; the hexane and chloroform fraction were 57.98 ± 1.97 and 71.37 ± 1.87, respectively, at this dose (Figure 6).

### 3.6. Reducing Potential

In Figure 7, it is revealed that the highest absorbance of the *S. mangifera* fractions was found for the ethanolic fraction (0.793 ± 0.03 nm) at 140 mg/mL, whereas the values for the hexane and chloroform fractions were 0.493 ± 0.05 nm and 0.598 ± 0.05 nm, respectively, at a concentration of 140 mg/mL.

### 3.7. Albumin Denaturation

The ethanolic fraction of *S. mangifera* significantly (*p <* 0.05) showed the greatest anti-denaturation effect for protein (89.57 ± 1.49 µg/mL), whereas the hexane and chloroform fractions had values of 57.61 ± 1.93 and 67.31 ± 1.39, respectively, when compared with acetylsalicylic acid (98.76 ± 1.93 µg/mL) at a concentration of 500 µg/mL (Figure 8).

### 3.8. Protease Inhibitor

A significant proteinase inhibitory efficacy (*p* < 0.05) was shown for the ethanolic fraction of *S. mangifera* (81.49 ± 2.14 µg/mL), whereas the hexane and chloroform fractions had values of 63.59 ± 1.97 µg/mL and 71.31 ± 1.49 µg/mL, respectively, and did not show significant (*p* > 0.05) action when compared with acetylsalicylic acid (98.76 ± 1.93) at a concentration of 500 µg/mL (Figure 9).

### 3.9. Anti-Arthritic Potential of S. mangifera against CFA-Induced Arthritis

When the CFA injected into the positive control group II, the animals’ paw diameter increased significantly (*p* < 0.01) from day 3 to day 28 when compared with the normal group animals. However, the percentage inhibition of the *S. mangifera* fraction at a dose of 400 mg/kg per day in the group VI animals was significantly (*p* < 0.05) decreased (83.84%) when compared with the positive control group II animals. Meanwhile, the percentage inhibition of the Hx-F and Chl-F fractions did not show a significant (*p* > 0.5) effect (53.07 and 60.74, respectively) on paw diameter when compared with the group II animals at this dose. However, the group III animals administered 100 mg/kg aspirin (standard) exhibited significantly (*p <* 0.01) reduced anti-arthritic potential (1.22 ± 1.16; *p* < 0.01) (Figure 10).

### 3.10. Arthritis Index

This study revealed a continuous and significant (*p <* 0.01) increase in the arthritic score in the group II animals compared to the group I animals. When the *S. mangifera* extract fraction was orally administered at a dose of 400 mg/kg; the group VI animals exhibited a significantly (*p <* 0.05) reduced (0.7± 0.27) arthritic score when compared with the group II animals, whereas the group IV and V animals did not show a significantly (*p >* 0.01) reduced (1.7 ± 0.11 1.6 ± 0.63) arthritis score. However, the group III animals administered 100 mg/kg aspirin (standard) exhibited significantly (*p <* 0.01) reduced (0.6 ± 0.18) arthritis score, similar to the group II animals (Figure 11).

## 4. Discussion

To obtain insights into the mechanism of action of the major constituents, beta amyrin, beta sitosterol, and oleanolic acid [34], they were docked into the active site of COX-1, COX-2, and TNF-α by Glide XP docking. β-Amyrin, β-sitosterol, and oleonolic acid showed excellent binding affinity with pro-inflammatory cytokine TNF-α. The co-crystallised ligand SPD-304 showed similar binding to that in the crystal structure, authenticating our docking protocol. SPD-304 was the first TNF-α antagonist to be identified, in 2005, and possesses a K_d_ value of 5.36 μM [24]. β-Amyrin, β-sitosterol, oleonolic acid and SPD-304 share high levels of similarity in their interactions with TNF-α in their corresponding bound states. A set of residues, consisting of Leu57A, Tyr59A, Gly121A, Tyr151A, Tyr59B, Tyr119B and Gly121B, is involved in hydrophobic interactions with β-amyrin, β-sitosterol, and oleonolic acid. These residues have been reported to be critical for the binding of the TNF-α antagonists [35].

Interestingly, for the rapid screening of vast numbers of bioactive natural compounds, in silico high-throughput ADME analysis and molecular docking screening is one excellent choice for quickly meeting this demand. These computer-aided techniques are used to explain or predict the toxicological and pharmacological effects of drugs; such studies reduce the cost and time required for experiments, as the drug discovery process is typically time-consuming and involves a huge amount of investment, and failure at any stage of bioactive drug development can lead to huge losses for an organisation. Notably, ADME study provides substantial information about the selected compounds with respect to the pharmacokinetic and toxicity profiles of compounds from plants, which can help in future drug development [36].

TNF alpha is responsible for inflammation when injury or other biological or mechanical processes occur. In the inflammation process, macrophage cells produce cytokinin and other proinflammatory agents in the body [37]. Nature provides numerous herbal plants that contain important bioactive phytoconstituents that can play a significant role in treatment or can be used as anti-inflammatory drugs, which has been helpful in the discovery of numerous anti-inflammatory agents [38]. Medications that are used for prevention and inhibition/blocking effects in preliminary reactions are set in a biological model by the leading cause, thus constraining established inflammation, and as a consequence, showing the established inflammation [39].

The ethanolic fraction of *S. mangifera* has the highest percentage of polyphenolic and flavonoidal compounds. DPPH (α, α-diphenyl-β-picrylhydrazyl) is composed of stable free-radical molecules, its free radical scavenging potential is based on its electron transfer ability; when mixed with fractions, it contributes a hydrogen atom, thus changing the colour of the fractions from purple to yellow as a result of the formation of diphenyl picrylhydrazine molecule. The grade of the stain indicates the scavenging potential of the fraction [40].

Acharya et al. suggested that the presence of reductones exerts the reducing powers of the extracts by contravention of the free radical chain, followed by the provision of a hydrogen atom [41]. The reducing ability depends on the presence of free radical molecules in the extracts or compound, which exert scavenging activity by giving a hydrogen atom and inhibiting the free radical chain in the system [42]. The naturally occurring phenolic composites contribute advantageous health properties by decreasing the free radical oxygen species [43].

Sadia et al. reported that plants have phenolic, polyphenolic and flavonoidal phytoconstituents, which exert scavenging activity owing to their redox properties. This allows them to react as reducing agents, hydrogen donors and singlet oxygen quenchers [44].

Modi et al. (2019) reported that the mechanism of protein denaturation changes the tertiary and secondary structures of protein due to alteration in electrostatic forces due to hydrophobia, disulphide bonds, and hydrogen [45], and some chemical and physical agents, like alcohol, acetone, acids, alkalies, heavy metal salts, dyes [46], heat, light and pressure, can also alter the protein denaturation. Mizushima and Kobayashi, in 1968, reported that the inflammatory drugs salicylic acid, phenylbutazone, etc., had a dose-dependent impact on thermally induced protein denaturation. Similarly, the protein denaturation ability of *S. mangifera* fractions has also been shown to be concentration dependent [47]. At the site of inflammation, the ethanolic fraction of *S. mangifera* probably reduces the release of the lysosomal content of neutrophils. These neutrophil constituents may include proteinases and bactericidal enzymes, whereby extracellular release may cause further inflammation and damage to the tissue [48]. Protein denaturation is a major factor for RA, causing the generation of auto antigens on a different order, and there is a linkage between hydrophobic electrostatic and disulphide denaturation [49]. In the present study, the in vitro assay of protein denaturation demonstrated the inhibition of protein denaturation in *S. mangifera* ethanolic fractions. According to this, the ethanolic fraction of *S. mangifera* could be helpful for protection against protein denaturation and auto antigen production, which are responsible for RA. The findings of this study are reported on the basis of protein denaturation assays using fresh hen and bovine serum albumin. These findings are in contrast with previous predictive research in which the extracts and standard revealed inhibition of protein denaturation in a dose-dependent manner [50].

The leukocyte proteinase has a significant role in the enhancement of tissue damage through inflammatory responses, and a substantial level of defence is conferred by proteinase inhibitors [51].

Proteinases play a significant role in the progression of tissue damage throughout the inflammatory processes. Several researchers have suggested that a significant level of protection is provided by the proteinase inhibitors [52]. It has already been mentioned in the literature that flavonoids isolated from plant sources possess significant antioxidant and anti-inflammatory activities [53].

The allopathic system of medicine can produce harmful side effects in living systems. The various anti-inflammatory (Ibuprofen, etoricoxib, corticosteroids, etc.) and antirheumatic drugs like methotrexate and hydroxyl chloroquine suphasalazine play important roles in life-threatening side effects [54]. In western and eastern countries, the use of herbal medicine is in extremely high demand by the public due to its compatibility with other products, and its lack of life-threating side effects or complaints [55]. Various studies have documented that herbal medicine can play an important role in the management of rheumatoid arthritis [54].

Therapeutic effects include anti-arthritic potential and antioxidant properties, and therefore the antioxidant actions of the *S. mangifera* fractions were assessed using the two parameters DPPH and reducing power-scavenging methods in order to maintain the equilibrium between free radical and antioxidants. Disturbances in this equilibrium destroy the cellular component and cause oxidative stress due to the generation of free radical species. Many diseases, such as inflammatory disorders and arthritic disease, are the result of oxidative stress, arising from free radical species. These reactive oxygen species react with organic supermolecules like lipids, proteins and DNA in the cells of living organism and produce harmful effects [56]. Herbal plants have many bioactive phytoconstituents, including flavonoids, phenolics, lignins, etc., which have the capability of fighting or scavenging the free radicals and providing protection against free radicals and reactive oxygen species [57].

CFA administered in the experimental rodents caused inflammation, mainly in the form of joint remodelling, synovial membrane infiltration and severe inflammation. These features resemble those of rheumatoid arthritis [58]. CFA is a commonly chosen animal model for evaluating inflammation, rheumatoid arthritis and autoimmune ailments in the context of herbal drugs. CFA-induced arthritis can be categorised into many phases, starting with the induction phase (no validation), through to early synovitis, followed my synovitis with devastation of joints [31]. In the early stages of arthritis, inflammation develops due to the release of prostaglandins, whereas when auto antibodies are produced, a secondary stage develops. The production of inflammatory mediators like IL-6 and TNF-α show a critical role in the advancement of joint disorder, bone distortion, pain, and tenderness [59].

The reduction in paw diameter indicated reduction in inflammatory mediators, suggesting that the drug has anti-inflammatory effects in CFA-induced arthritis [60]. In this study, the group II (toxic control) animals exhibited continued swelling for 28 days due to cellular incursion, with constant oedema occurring, in contrast to the normal group I animals. However, in the ethanolic-fraction-treated group VI animals, maximum swelling was noticed at up to 7 days, with a reduction in inflammation beginning from day 8, in contrast with the group II animals. In this study, we found that these parameters were suitable for measuring the efficiency of anti-arthritic medications [35].

In this study, *S. mangifera* demonstrated a reduction in joint inflammation and arthritis index, meaning that the immunosuppressive and the anti-inflammatory properties are differentiated [59]. The *S. mangifera* alcoholic fraction resulted in a comparative decline in the index and protection against morphological alterations. These changes were due to a reduction in swelling, while redness and secondary growth were inhibited due to its anti-arthritic potential [61]. The use of herbal drugs is met with better tolerance, and with fewer adverse effects. Drug discovery in the field of phytoconstituents obtained from natural sources could be advantageous for the pharmaceutical market as well as patient compliance.

## 5. Conclusions and Future Perspective

There are a number of plants that contain phytochemicals, which play an important role in human life and provide beneficial effects. The plant *S. mangifera* is a well-known plant possessing different phytoconstituents, e.g., β sitosterol, β amyrin and oleonolic acid. The ethanolic fraction of the plant is a good adjuvant in the present armamentarium. Therefore, it could be a beneficial agent for the management of inflammation and arthritis. The results of docking studies support the in vivo and in vitro data, showing that the active constituents of the plant interface with chain A and B of TNF-α. The ADME studies show that among the phytoconstituents, gallic acid shows the best ADME properties, and the compound did not cross the blood–brain barrier. The in vivo data reveal the beneficial effects of the alcoholic fraction of *S. mangifera* with respect to inhibiting the free radical scavenging assay, and inhibition of protein denaturation by inhibiting the proteinase enzymes. This evidence shows that the alcoholic fraction of *S. mangifera* has good anti-arthritic and anti-inflammatory potential, and thus could be used in arthritis management and as a potent novel drug delivery agent. By using phytoconstituents for the management of disease, we can avoid the side effects of synthetic drugs.

## Figures and Tables

**Figure 1 plants-10-00825-f001:**
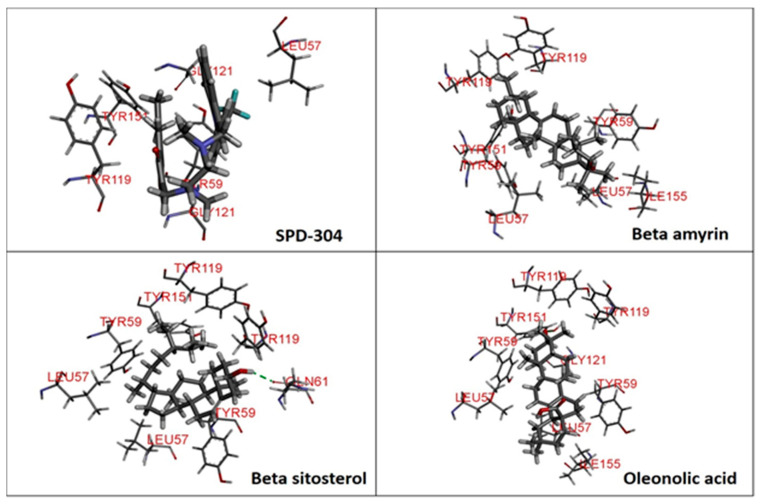
2D representation of docked conformation of TNF-α with ligands obtained after glide XP docking. Green dashed lines represent conventional hydrogen bonds with the interacting amino acid residues. Pink lines indicate hydrophobic (alkyl–alkyl, π–alkyl or π–π) interactions.

**Figure 2 plants-10-00825-f002:**
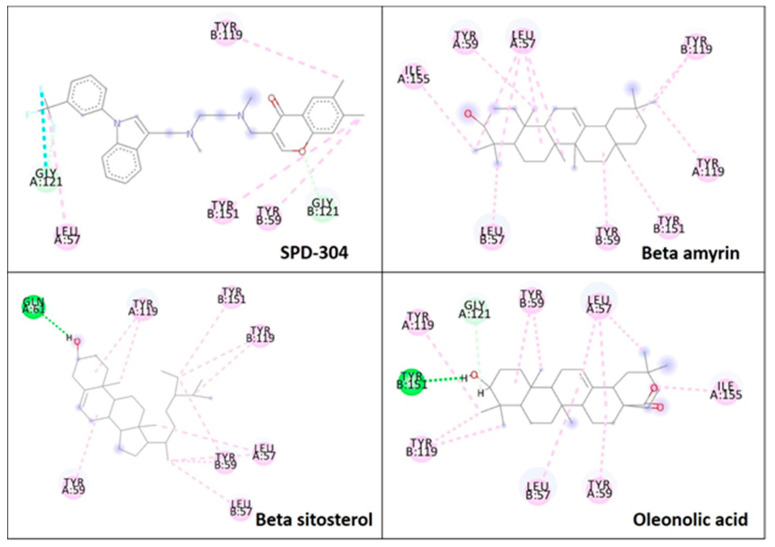
3D representation of docked conformation of TNF-α with ligands obtained after Glide XP docking.

**Figure 3 plants-10-00825-f003:**
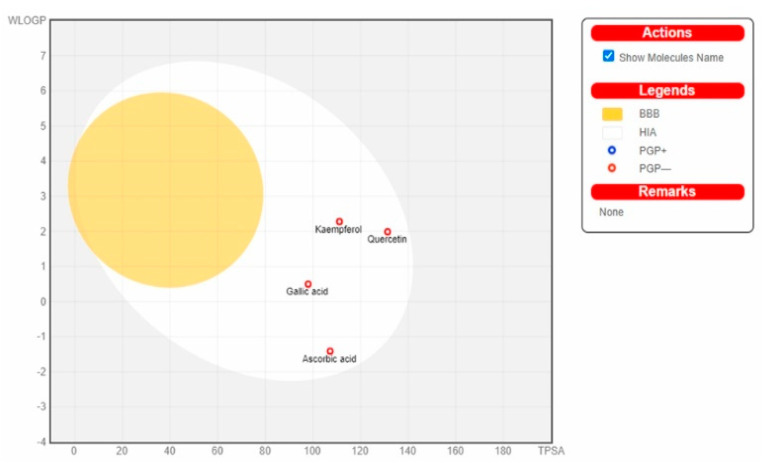
The figure shows that no compounds cross blood–brain barrier. Among these compounds, the compounds gallic acid and caffeic exhibit the best ADME properties.

**Figure 4 plants-10-00825-f004:**
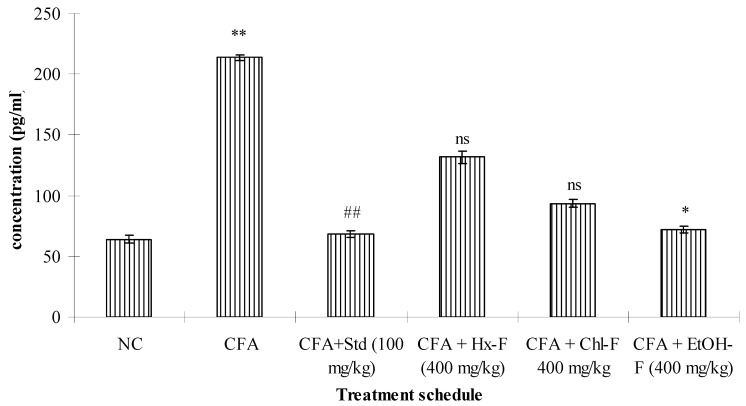
Effect of *S. mangifera* ethanolic fraction on TNF-α. Results are expressed as mean ± SEM (*n* = 6) and analysed by one-way ANOVA followed by Dunnet’s test. ** *p* < 0.01 = significant when compared with group I. ^##^
*p <* 0.01, * *p <* 0.05 = significant when compared with group II. ^ns^
*p* > 0.05 = non-significant when compared with group II.

**Figure 5 plants-10-00825-f005:**
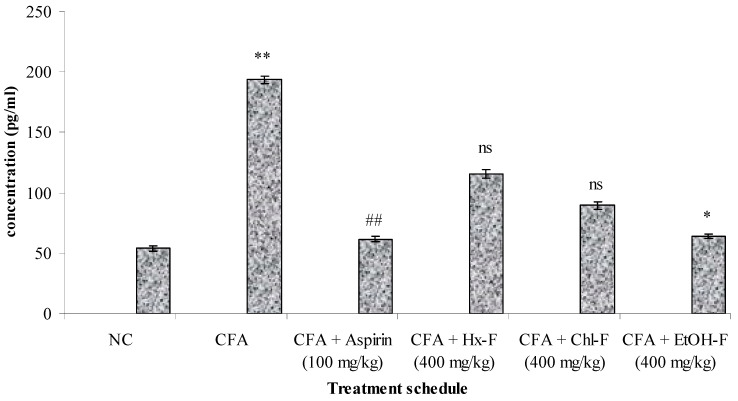
Effect of *S. mangifera* ethanolic fraction on IL-6. Results are expressed as mean ± SEM (*n* = 6) and analysed by one-way ANOVA followed by Dunnet’s test. ** *p* < 0.01 = significant when compared with group I. ^##^
*p <* 0.01, * *p <* 0.05 = significant when compared with group II. ^ns^
*p* > 0.05 = non-significant when compared with group II.

**Figure 6 plants-10-00825-f006:**
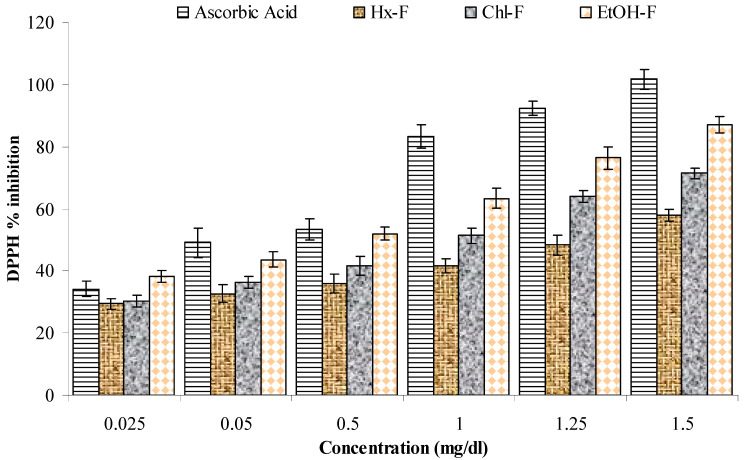
Percentage inhibition of DPPH scavenging activity of *S. mangifera* fruit fractions. Data are presented as the mean value ± SEM (*n* = 3).

**Figure 7 plants-10-00825-f007:**
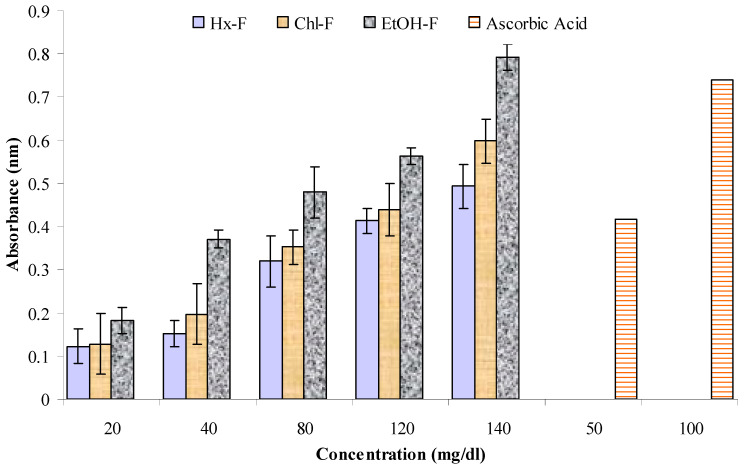
Reduced potential scavenging activity of different fractions of *S. mangifera* fruit extract.

**Figure 8 plants-10-00825-f008:**
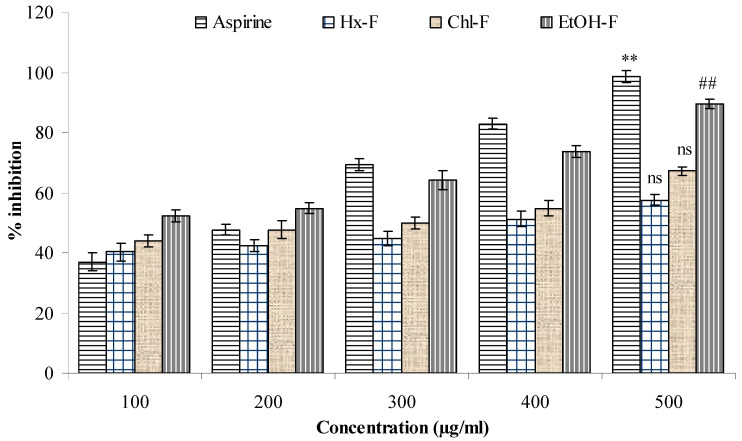
Effect of *S. mangifera* ethanolic fraction against protein denaturation using egg albumin. Data are expressed as means ± SEM (*n* = 3), with a significance test for comparison with aspirin using ANOVA followed by Dunnet’s ‘*t*’ test. ** *p <* 0.01, ^##^
*p* < 0.05 and ^ns^
*p* > 0.05: non-significant.

**Figure 9 plants-10-00825-f009:**
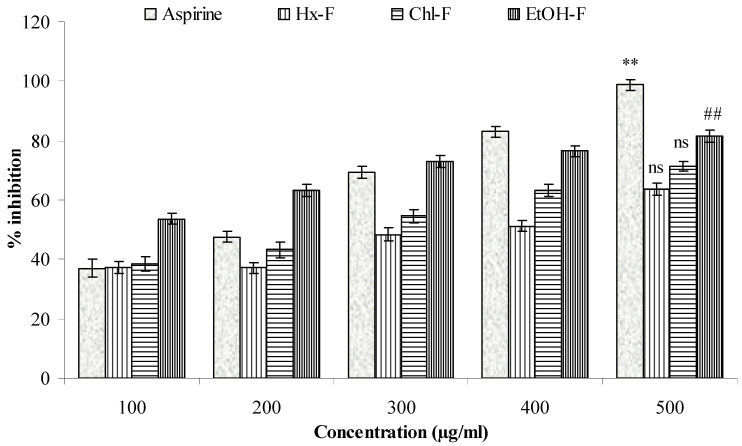
Effect of *S. mangifera* ethanolic fraction against protein denaturation using egg albumin. Data are expressed as means ± SEM (*n* = 3), with a significance test for comparison with aspirin using ANOVA followed by Dunnet’s ‘*t*’ test. ** *p* < 0.01, ^##^
*p* < 0.5 and ^ns^
*p* > 0.05: non-significant.

**Figure 10 plants-10-00825-f010:**
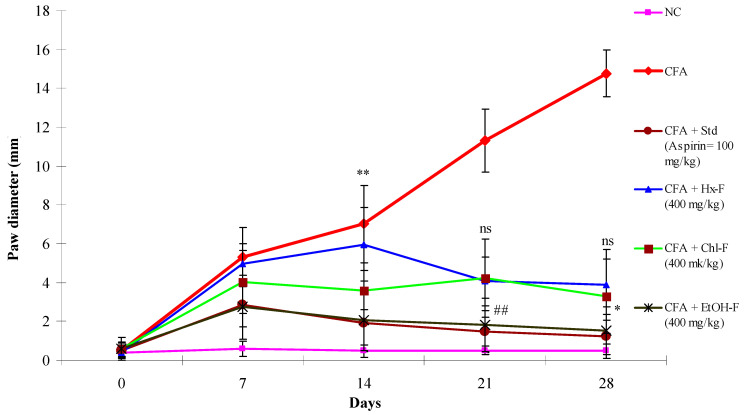
Effects of *S. mangifera* ethanolic fraction on CFA-induced arthritis in mice. All values are expressed as Mean = SEM (*n* = 6). ** *p <* 0.01= significant when compared with group I. ^##^
*p* < 0.01 and * *p <* 0.05 = significant when compared with group II. ^ns^
*p* > 0.05: non-significant when compared with group II.

**Figure 11 plants-10-00825-f011:**
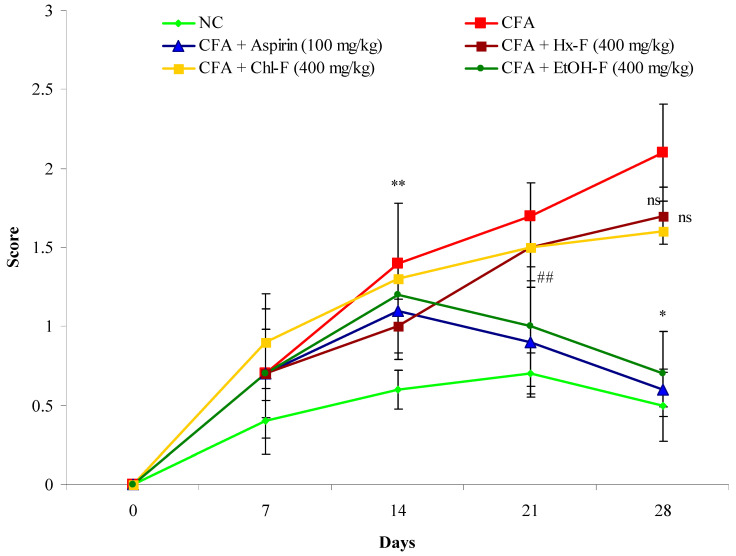
Effects of *S. mangifera* ethanolic fraction on CFA-induced arthritis in mice. All values are expressed as Mean = SEM (*n* = 6), ** *p <* 0.01 = significant when compared with group I. ^##^
*p* < 0.01 and * *p <* 0.05 = significant when compared with group II. ^ns^
*p* > 0.05: non-significant when compared with group II.

**Table 1 plants-10-00825-t001:** Summary of the selected constituents and the reference compound (SPD-304); chemical structures, Glide XP docking scores, hydrogen bond interactions and close contact residues.

Ligands	Chemical Structure	Binding Energy Score (Kcal/Mol)	Interaction (H-Bond)	Interaction (Hydrophobic)
SPD-304	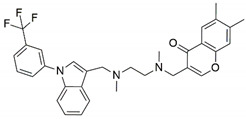	−8.137	Gly A: 121	Leu A: 57, Tyr B: 59, Tyr B: 119, Gly B: 121, Tyr B: 151
β-amyrin	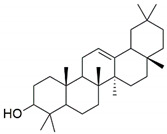	−7.517	-	Leu A: 57, Tyr A: 59, Tyr A: 119, Ile A: 155, Leu B: 57, Tyr B: 59, Tyr B: 119, Tyr B: 151
β-sitosterol	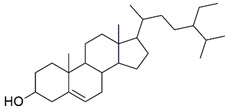	−9.190	Gln A: 61	Leu A: 57, Tyr A: 59, Tyr A: 119, Leu B: 57, Tyr B: 59, Tyr B: 119, Tyr B: 151
Oleonolic acid	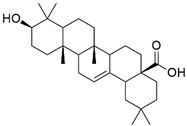	−8.178	Gly A: 121 Tyr B: 151	Leu A: 57, Tyr A: 59, Tyr A: 119, Ile A: 155, Leu B: 57, Tyr B: 59, Tyr B: 119

**Table 2 plants-10-00825-t002:** Pharmacokinetic parameters of phytoconstituents of *S. mangifera* fruits.

Molecule	MW	Ali Class	GI Absorption	BBB Permeant	Log Kp (cm/s)	Lipinski #Violations
Gallic acid	170.12	Soluble	High	No	−6.84	0
Kaempferol	286.24	Soluble	High	No	−6.70	0
Quercetin	302.24	Soluble	High	No	−7.05	0
Ascorbic acid	176.12	Very soluble	High	No	−8.54	0

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
