# Peer review of "Anti-Arthritic and Anti-Inflammatory Potential of Spondias mangifera Extract Fractions: An In Silico, In Vitro and In Vivo Approach"

_plants, 2021, doi:10.3390/plants10050825_

Round 1

Reviewer 1 Report

The paper looks interesting by means of both scientific soundness as well as some potential application. The use of materials from folk medicine with full understanding the role of particular constituents broaden the knowledge and give a chance for many patients to get better and fight the diseases.

Additionally the experiments are well designed  (except the drying of fruits see point 1 in the list below) allowing to get as much scientific information as it is needed to reach the main goal.

The discussion section is well prepared however conclusions should be redesigned (point 4). According to that I would like to designate the paper as needed minor revision only. A short list of my suggestion is presented below.

  1. Please explain a method for fruits drying (including all parameters needed)
  2. Please add reference showing that beta amyrin, beta sitosterol, and oleanolic acid are main bioactive constituent of fruits
  3. The are several typos in the text. Please scan the text carefully in order to fix it (eg. Line 382 or EoH vs. EtOH)
  4. Please rewrite conclusions section in order to give not a general picture but rather some details.

Author Response

Reply to Reviewer 1

  1. Please explain a method for fruits drying (including all parameters needed)

Reply: The S. mangifera fruits were collected and dried; for the drying process they were scattered on a dry open area at room temperature. After air drying these fruits were cut into four pieces by the means of a sharp knife. And again, dried in oven at 40 - 45 ℃ for a period of 2-3 days until constant weight. The dried fruits were made into coarse powder with the help of grinder and now this powder materials were defatted with petroleum ether and further macerated with methanol for 72 hours with occasional shaking.

  1. Please add reference showing that beta amyrin, beta sitosterol, and oleanolic acid are main bioactive constituent of fruits

Reply: Arif, M.; Rahman, M.A.; Imran, M.; Khalid, M.; Khushtar, M. An insight of Spondias mangifera Willd: an underutilized medicinal plant with immense nutraceutical and therapeutic potentials. Int J Res Pharm Sci 2015; 6, 17-26. (Added in manuscript as citation number: 34).

  1. The are several typos in the text. Please scan the text carefully in order to fix it (eg. Line 382 or EoH vs. EtOH)

Reply: Thank you for the keen observation and all corrections are done in the revised manuscripts.

  1. Please rewrite conclusions section in order to give not a general picture but rather some details.

Reply: There are number of plants having phytochemicals which play an important role in human life and provide a beneficial effect. The plant S. mangifera is a well-known plant having different phytoconstituents viz., β sitosterol, β amyrin and oleonolic acid. The ethanolic fraction of plant possesses a good adjuvant in the present armamentarium. So that It can be a beneficial agent for the management of inflammation and arthritis. The results of docking studies support the in-vivo and in-vitro data; that shows that the active constituents of the plant interface with chain A and B of TNF-α. The ADME studies shows that among all phytoconstituents gallic acid shows better ADME property and the compound of the plant did not cross blood brain barrier. The in-vivo data reveals the beneficial effect of the alcoholic fraction of S. mangifera by inhibiting the free radical scavenging assay, inhibition of protein denaturation by inhibiting the proteinase enzymes. These evident showed the alcoholic fraction of S. mangifera has good antiarthritic and anti-inflammatory- potential hence it could be used in arthritis management and can be used as a potent target as a novel drug delivery agent. By the means of using phytoconstituents for the management of disease we can avoid the side effects of synthetic drugs.

Reviewer 2 Report

This paper's aim was to present the anti-arthritis and anti-inflammatory potential of Spondias mangifera fruits extract. The manuscript fits within the scope of the journal. The manuscript is interesting and well organized.  The title is clear and it is adequate to the content of the article.  Some revisions are necessary to improve the clarity of the presentation.

I have some recommendations for authors:

- First and foremost, the manuscript needs editing for both grammar and spelling. It was difficult at times, to overcome the grammatical errors found in the document.

- Please include the aim of the study at the end of the introduction.

- Please include citations for all paragraphs: eg:  “The characteristic feature of RA involve is join pain, immobility and malformation.”

“The management of RA is mainly the using of nonsteroidal anti-inflammatory drugs (NSAIDS) like derivatives of indole-acetic acid, ibuprofen, aspirin, naproxen are members of this category of drugs, but it’s managed in short term duration.”

“The arthritic and antiinflammatory action of NSAID is attributed to it cyclooxygenase (COX-1 & 2) inhibition, and also inhibited the pro cytokinin’s (IL1, IL-6 TNF-alpha etc) to cure the arthritic disease.”

“The green fruit is useful in bilious dyspepsia, and fruits powder have anti-tubercular properties, also used as astringent, refrigerant, tonic and used for management of rheumatic arthritis and myalgia respectively.”

Please improve quality of all graphics and figures. From my point of view you used a lot of colors, which tire the reader.

Please use italic for scientific name in all text (eg. abstract, figure titles…)

Please include some information about the degree of novelty/originality of the results. What are the future applications? What are the next research directions? Please detail in discussions and conclusion sections.

The author’s work on discussing achieved results is appreciated. The revisions are necessary to improve the clarity of the presentation and needed to make convincing scientific arguments.

Author Response

Reply to Reviewer 2

  1. First and foremost, the manuscript needs editing for both grammar and spelling. It was difficult at times, to overcome the grammatical errors found in the document.

Reply: All suggested correction is done in revised manuscript.

  1. Please include the aim of the study at the end of the introduction.

Reply: The present study was designed to find out the pharmacokinetics and pharmaco-dynamic properties of the Spondias mangifera. Among all phytochemicals β sitosterol, β amyrin and oleonolic acid are active. Keeping in mind the special considerations to find out the anti-inflammatory and anti-arthritic actions of the plant. First the active constituents are tested for the docking study to find out the binding affinity and then fur-there tested for pharmacokinetics study lastly ethanolic fraction of the plant extract was used for the estimation of pharmacological activity against inflammation and CFA induced arthritis in animal model. 

  1. Please include citations for all paragraphs:

Reply: “The characteristic feature of RA involve is join pain, immobility and malformation.” Cited as 5 reference in revised manuscript.

The management of RA is mainly the using of nonsteroidal anti-inflammatory drugs (NSAIDS) like Indomethacin, ibuprofen, aspirin, naproxen are members of this category of drugs, but it’s managed in short term duration.” Cited as 6 reference in revised manuscript.

“The arthritic and antiinflammatory action of NSAID is attributed to it cyclooxygenase (COX-1 & 2) inhibition, and also inhibited the pro cytokinin’s (IL1, IL-6 TNF-alpha etc) to cure the arthritic disease.” Cited as 7 reference in revised manuscript.

“The green fruit is useful in bilious, dyspepsia, and fruits powder have anti-tubercular properties, also used as astringent, refrigerant, tonic and used for management of rheumatic arthritis and myalgia respectively.” Cited as 12 reference in revised manuscript.

  1. Please improve quality of all graphics and figures. From my point of view, you used a lot of colors, which tire the reader.

Reply: As per the suggestion we have change the color of the images. All the revised images are added in the manuscript.

  1. Please use italic for scientific name in all text (eg. abstract, figure titles…)

Reply: As per the suggestion we have changed all the scientific name/ Botanical name into italic font.

  1. Please include some information about the degree of novelty/originality of the results.

Reply: To the best of our knowledge this is the first study which includes the pharmacokinetics, dynamics, in-silico, in-vitro and in-vivo studies on the plant Spondias Mangifera extract fraction. That shows the probable action of the plant as anti-inflammatory and an-ti-arthritic. The outcomes of the results can be proved by several aspects:

  • Molecular docking
  • Server based ADME analysis
  • Cyclooxygenase assay
  1. What are the future applications? What are the next research directions? Please detail in discussions and conclusion sections.

Reply:  There are number of plants having phytochemicals which play an important role in human life and provide a beneficial effect. The plant S. mangifera is a well-known plant having different phytoconstituents viz., β sitosterol, β amyrin and oleonolic acid. The ethanolic fraction of plant possesses a good adjuvant in the present armamentarium. So that It can be a beneficial agent for the management of inflammation and arthritis. The results of docking studies support the in-vivo and in-vitro data; that shows that the active constituents of the plant interface with chain A and B of TNF-α. The ADME studies shows that among all phytoconstituents gallic acid shows better ADME property and the compound of the plant did not cross blood brain barrier. The in-vivo data reveals the beneficial effect of the alcoholic fraction of S. mangifera by inhibiting the free radical scavenging assay, inhibition of protein denaturation by inhibiting the proteinase enzymes. These evident showed the alcoholic fraction of S. mangifera has good antiarthritic and anti-inflammatory- potential hence it could be used in arthritis management and can be used as a potent target as a novel drug delivery agent. By the means of using phytoconstituents for the management of disease we can avoid the side effects of synthetic drugs.

Round 2

Reviewer 2 Report

The authors improved the manuscript and made the requested changes.